# Seasonal Enhancement of Nitrogen Removal on Domestic Wastewater Treatment Performance by Partially Saturated and Saturated Hybrid Constructed Wetland



**José Contreras [1], Daniela López [1,2], Gloria Gómez [1] and Gladys Vidal [1,*]**

1     Engineering and Environmental Biotechnology Group, Environmental Science Faculty & Center EULA-Chile, Universidad de Concepción, Concepción 4070386, Chile; contreraspincheira1@gmail.com (J.C.); spdanielita@gmail.com (D.L.); gloriagomezosorio@gmail.com (G.G.)

2     Facultad de Ingeniería y Ciencias, Universidad Adolfo Ibáñez, Avda. Padre Hurtado 750, Viña del Mar 2562340, Chile

\*     Correspondence: glvidal@udec.cl; Tel.: +56-41-2204067; Fax: +56-41-2207076

**Abstract:** The aim of this study is to evaluate seasonal enhancement of nitrogen removal on domestic wastewater treatment performance by partially saturated and saturated HBCWs. To achieve this, two HBCWs consisting of a vertical subsurface flow constructed wetland, followed by a horizontal subsurface flow constructed wetland (VSSF-HSSF) were evaluated. Two saturation levels were used: (a) partially saturated HB1:VSSF1 (0.6 m)-HSSF1 (0.15 m), (b) saturated HB2: VSSF2 (0.8 m)-HSSF2 (0.25 m). Each unit was planted with *Schoenoplectus californicus* and was operated for 297 days. The removal efficiencies in HB1 and HB2 were above 70%, 86%, 77% and 55% for chemical oxygen demand (COD), total suspended solids (TSS), nitrogen as ammonium ($NH_4^+$-N), and total nitrogen (TN), respectively. For VSSF, a higher level of saturation (from 0.6 to 0.8 m) meant a decrease of 17% in the TN removal efficiencies, and for HSSF, an increase from 0.15 to 0.25 m of saturation meant a decrease of 11 and 10% in the $NH_4^+$-N and TN removal efficiencies, respectively. Thus, the increase of saturation level in HBCWs reduces the transformation and/or removal of components of the wastewaters to be treated, particularly nitrogen. Through this research, the possibility of optimizing the transformation of nitrogen with partially saturated hybrids can be examined.

**Keywords:** hybrid constructed wetland; saturation level; organic matter; ammonia





## 1. Introduction

Hybrid constructed wetlands (HBCWs) are based on the sequential utilization of several constructed wetlands (CWs), allowing the combination of each system advantage. Moreover, several configurations have been used, VSSF + VSSF, HSSF + HSSF and VSSF + HSSF, the latter being the most commonly used to treat municipal and industrial wastewater [1–5]. VSSFs are characterized by an oxygen transfer of 28–100 $gO_2/m^2 \cdot d$, maintaining aerobic conditions and allowing processes such as nitrification with elimination rates of 0.01–4.17 $gNH_4^+/m^2 \cdot d$ [6–9]. In the case of HSSF, oxygen transfer occurs at 0.3–3.2 $gO_2/m^2 \cdot d$, maintaining anaerobic conditions and allowing processes such as denitrification if nitrate is present with elimination rates of 0.47–0.83 $gNO_3^-/m^2 \cdot d$ [6,8–10]. Several studies have reported that HBCWs present removal efficiencies for COD (58–97.6%), TSS (94–96%), TN (62.8–97%) and $NH_4^+$-N (81.1–99.3%) [7,11–14].

Among the design parameters, it has been stated that the wastewater saturation level impacts the different processes that occur in the HBCWs. [15,16]. Various authors have investigated the influence of partial saturation on the component's transformation in HSSF or VSSF separately, evidencing the potentialities to further improve the elimination efficiencies of these components. That is how Sanchez-Ramos et al. [17] determined higher removal efficiencies (94% vs. 83% for COD and 74% vs. 32% for $NH_4^+$-N) in an HSSF

under different saturation levels (0.25 and 0.50 m) than a system that presented a lower saturation level.

Moreover, García et al. [15] determined in a partially saturated HSSF (0.27 m) and in a saturated HSSF (0.50 m) that the best removal efficiencies for COD and $NH_4^+$-N were in the partially saturated system with values up to 23.83 and 69.35% higher than those in a saturated HSSF. Huang et al. [18] used a VSSF with different saturation levels (0.05, 0.30, 0.45 and 0.60 m) to improve nitrogen removal. The results indicated that the modification of the saturation level significantly affected the transformation of nitrogen, and the optimal level of saturation was achieved at 0.45 m, where the authors reported that transformation/removal efficiencies of TN and $NH_4^+$-N of 76.65 and 82.91% enabled the generation of other processes, such as anammox. Furthermore, Saeed and Sun [19] employed a VSSF under unsaturated and partially saturated conditions with saturation levels of 0.14, 0.29 and 0.43 m, respectively, and conducted a comparative analysis on contaminant removal. They observed that VSSF with a lower saturation level had higher removal efficiencies of $BOD_5$ (47%), TN (52%) and $NH_4^+$-N (53%), as it presented an aerobic environment (2.2 mg/L), favoring processes such as nitrification with mean removal rates of 1.30 $gNH_4^+/m^2 \cdot d$.

Specifically, partial saturation in CWs allows the development of an anoxic/anaerobic zone inside the saturated support medium in VSSF and the penetration of atmospheric oxygen toward the saturated zone through overlaid unsaturated bulk media volume without altering commonly practiced overall main mean depth of traditional fully saturated HF wetlands [20]. Despite the advances in research that have been made regarding partial saturation and single-stage CWs, investigations about transformation of nitrogen with partially saturated HBCWs is very limited. Given all that has been mentioned thus far, the main objective of this study is to evaluate the saturation level in the in the treatment of municipal wastewater through hybrid constructed wetlands. Specifically, we will evaluate the saturation of wastewater level in the transformation of nitrogen through hybrid constructed wetlands.

## 2. Materials and Methods

### 2.1. Hybrid Constructed Wetlands

The wastewater influent was collected after primary treatment from the wastewater treatment plant in Hualqui, Concepción Province, Biobío Region (Chile) (36°59′26.93″ from south latitude and 72°56′47.23″ from west latitude), which can serve 20,000 inhabitants [21,22]. The influent was transported and stored in 20 L tanks and refrigerated at 4 °C in the dark [23].

Two hybrid constructed systems (HBCWs) were used to scale at a laboratory at Universidad de Concepción. The systems were implemented in a greenhouse-type laboratory with semicontrolled temperatures with between 1–20 °C corresponding to a temperate maritime climate with Mediterranean influence. These HBCWs consisted of a vertical subsurface flow constructed wetland, followed by a horizontal subsurface flow constructed wetland (VSSF-HSSF). In addition, different saturation levels were used. For the VSSF, a partially saturated model (0.6 m) and a saturated model (0.8 m) were employed. Similarly, for the HSSF, partially saturated (0.15 m) and saturated models (0.25 m) were also employed. Therefore, the experimental units for the HBCWs consisted of (a) HB 1: VSSF 1 (0.6 m)-HSSF 1 (0.15 m) and (b) HB 2: VSSF 2 (0.8 m)-HSSF 2 (0.25 m). Each of the four units was planted with the macrophyte species (*Schoenoplectus californicus*). The VSSF and HSSF systems were inoculated with 4 and 8 seedlings distributed homogeneously, respectively.

Each system was fed discontinuously with 3.6–4.2 L/d for VSSF 1 and VSSF 2 and 1.1–1.7 L/d for HSSF 1 and HSSF 2, respectively. The VSSF had a hydraulic retention time (HTR) of 1 d, and the HSSF had an HRT of 7 d. The effluent was collected using pipeline systems located underneath each wetland. The main design and operational characteristics of the CW are shown in Table 1. Moreover, Figure 1 shows a schematic representation of the HBCWs.

**Table 1.** Main characteristics of a hybrid constructed wetland.

| Characteristics | Unit | VSSF | HSSF |
|---|---|---|---|
| Design parameters | | | |
| Surface area | m$^2$ | 0.025 | 0.17 |
| Average height | m | 0.85 | 0.30 |
| Water table height | m | 0.6–0.8 | 0.15–0.25 |
| Total volume | m$^3$ | 0.02 | 0.05 |
| Support medium | | | |
| Type | - | Sand/Gravel | Gravel |
| Size | mm | 1–4/19–25 | 19–25 |
| Porosity | - | 0.2/0.4 | 0.4 |
| Operation parameters | | | |
| HRT | d | 1 | 7 |
| HLR | L/d | 3.6–4.2 | 1.1–1.7 |

HRT: Hydraulic retention time; HLR: Hydraulic loading rate.

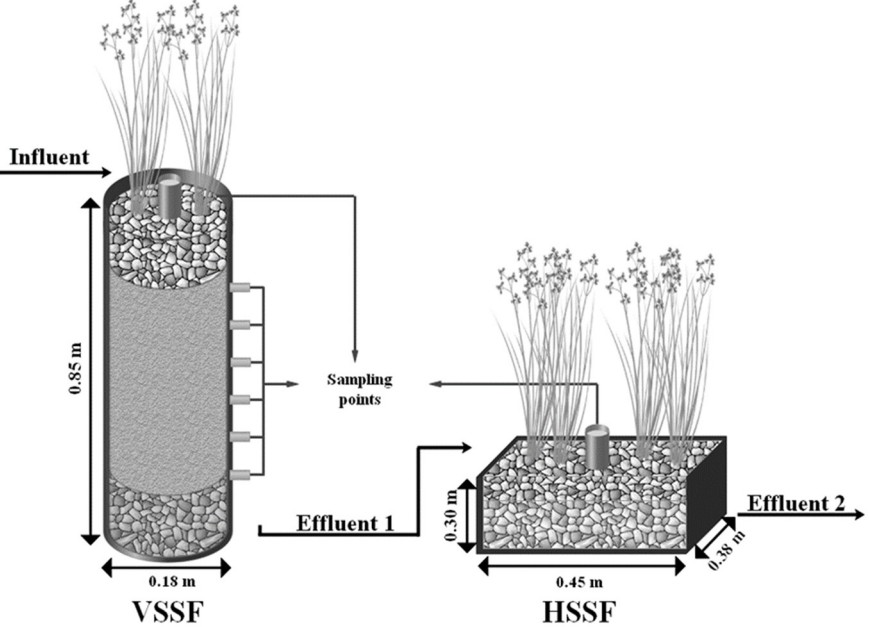

**Figure 1.** Schematic representation of hybrid constructed wetlands.

## 2.2. Sampling Strategy

During the operational period (a total of 297 days), which included summer, autumn, winter, and spring, the in situ parameters, including pH temperature, electrical conductivity, dissolved oxygen (DO) and oxide reduction potential (ORP), were measured weekly using the samplers located at the center of each wetland. Physicochemical parameters, including chemical oxygen demand (COD), total organic carbon (TOC), total suspended solids (TSS), volatile suspended solids (VSS), nitrate-nitrogen ($NO_2^-$-N), the nitrogen of the nitrate ($NO_3^-$-N) and ammonium nitrogen ($NH_4^+$-N), were measured weekly. Total nitrogen (TN) was measured monthly [22,24]. Removal efficiencies were determined according to the Equation (1) used by Tuttolomondo et al. [25].

$$R(\%) = ((Q_i \cdot C_i - Q_e \cdot C_e)/(Q_i \cdot C_i)) \cdot 100 \tag{1}$$

where $Q_i$ is the influent flow rate, $Q_e$ is the effluent flow rate, $C_i$ and $C_e$ are the influent and effluent pollutant concentration, respectively.

### 2.3. Analitycal Methods

To monitor the HBCWs, samples were taken of the influent and the output of each unit, which were filtered through a Whatman 0.45 mm membrane pore size. All protocols described in the standard methods were followed for the physicochemical parameters [26]. Combustion catalytic oxidation at 680 °C and detection by a nondispersive infrared sensor (NDIR) were used to determine TOC. The COD was determined using the colorimetric method (5210-B). The TSS and VSS were determined by the gravimetric method. The nutrients $NO_3^-$-N were determined by UV, $NO_2^-$-N and $NH_4^+$-N spectrophotometry, which were determined using the colorimetric method, and TN was determined by Spectroquant-Nova 60 (kits of Merck, Kenilworth, NJ, USA). For the in situ parameters, OAKTON portable multiparameter equipment (PC650-480485, Charleston, SC, USA) was used, and for DO, a portable oximeter (HANNA OXI 330i/set HI 9146-04, Villafranca, Italy) was used.

Statistical analysis was performed to compare the removal efficiencies of pollutants of the different CW units. Data were grouped, and the Shapiro–Wilk normality test was performed. A paired t-test was used if the data were normally distributed, and data without normal distribution were analyzed with a Wilcoxon test. The effect of the saturation level of the HBCWs was compared; if they were characterized by a normal distribution, an ANOVA test was performed, but if data were not normally distributed, the Kruskal–Wallis test was used to conduct the analysis. All statistical analyses were performed using the InfoStat statistical program with a significance level of 0.05 [27].

## 3. Results and Discussion

### 3.1. Influent Physicochemical Characterization

Table 2 shows the results of the physicochemical characterization of the influent throughout the duration of this study. The influent had an average COD value of 211.67 mg/L and a range of 127.37–401.17 mg/L, reflected in the variations in COD concentrations when considering municipal wastewater [28]. There were applied organic load rates (OLRs) in the range of 18.23–57.76 g/m²·d for HB 1 and 21.27–67.40 g/m²·d for HB 2. Similar results were obtained by García-Ávila et al. [29], who registered an influent with an average value of 222.44 mg/L for municipal wastewater. Likewise, Al-Ajalin et al. [30] reported influents with COD concentrations of 234.00 mg/L regarding municipal wastewater. Additionally, the highest concentrations of COD were recorded during autumn with an average of 296.94 mg/L. In contrast, the lowest concentrations of COD were recorded during spring with a mean of 178.51 mg/L. Thus, the highest variation between seasons was recorded during autumn and spring at 44%.

**Table 2.** Physicochemical characterization of the influent.

| | Concentration (mg/L) ± SD | | | | | | | |
|---|---|---|---|---|---|---|---|---|
| | Summer | | Autumn | | Winter | | Spring | |
| | Average ± SD | Range | Average ± SD | Range | Average ± SD | Range | Average ± SD | Range |
| COD | 200.67 ± 66.49 | 139.93–314.73 | 296.94 ± 66.75 | 193.93–401.17 | 183.53 ± 42.77 | 132.35–251.49 | 178.51 ± 69.48 | 127.37–257.61 |
| TOC | 45.72 ± 22.31 | 16.35–85.13 | 87.09 ± 21.41 | 52.45–116.00 | 49.64 ± 11.57 | 35.80–68.02 | 48.28 ± 18.73 | 34.45–69.68 |
| TSS | 98.96 ± 11.23 | 20.94–106.92 | 42.94 ± 12.67 | 25.20–61.00 | 38.25 ± 13.82 | 27.00–59.33 | 32.67 ± 2.52 | 30.00–35.00 |
| VSS | 24.25 ± 9.31 | 18.75–35.00 | 40.26 ± 11.97 | 23.20–58.00 | 33.25 ± 14.18 | 15.50–57.33 | 28.20 ± 1.08 | 27.00–29.10 |
| $NO_2^-$-N | 0.54 ± 0.04 | 0.51–0.57 | 0.42 ± 0.18 | 0.17–0.72 | 0.31 ± 0.05 | 0.25–0.49 | 0.36 ± 0.12 | 0.27–0.49 |
| $NO_3^-$-N | 1.69 ± 0.33 | 1.26–2.15 | 1.98 ± 0.53 | 1.34–2.86 | 1.22 ± 0.26 | 0.58–1.83 | 0.33 ± 0.07 | 0.28–0.38 |
| $NH_4^+$-N | 81.79 ± 12.44 | 66.34–96.38 | 103.66 ± 7.91 | 92.71–123.78 | 95.81 ± 9.90 | 81.73–106.03 | 97.23 ± 15.74 | 80.05–110.98 |
| TN | 105.00 ± 19.00 | 86.00–124.00 | 130.67 ± 5.69 | 126.00–137.00 | 137.67 ± 6.66 | 132.00–145.00 | 130.00 ± 2.83 | 128.00–132.00 |

COD: chemical oxygen demand; TOC: total organic carbon; TSS: total suspended solids; VSS: volatile suspended solids; $NO_2^-$-N: nitrite nitrogen; $NO_3^-$-N: nitrate nitrogen; $NH_4^+$-N: ammonium nitrogen; TN: total nitrogen.

The TSS presented a range of 32.67–98.96 mg/L with a mean of 53.20 mg/L. Higher concentrations were recorded during summer with a mean of 98.96 mg/L. Moreover, the highest observed variation was recorded between summer and spring at 66%. Similar results were reported by Zurita et al. [31] with a TSS average concentration value of 57.50 mg/L for domestic wastewater.

In terms of nitrogen, the TN recorded a concentration range from 86 to 145 mg/L with higher concentrations during winter (137.67 mg/L average) and up to 23.73% higher than during summer (105.00 mg/L average). $NH_4^+$-N had a concentration range from 66.34 to 123.78 mg/L, and a variation of 21.10% was registered in summer (81.79 mg/L average) and autumn (103.66 mg/L average). The applied nitrogen load rates (NLRs) were, on average, 12.80 and 15.01 g $NH_4^+$-N/$m^2$·d for VSSF 1 and VSSF 2, respectively. On the other hand, HSSF 1 and HSSF 2 registered applied nitrogen loads of 0.44 and 0.67 g $NH_4^+$-N/$m^2$·d, respectively. $NO_2^-$-N registered an average value of 0.40 mg/L with higher concentrations during summer (0.54 mg/L average) and lower concentrations during spring (0.25 mg/L on average). Regarding $NO_3^-$-N, a variation of 83.33% during autumn and spring was registered with a concentration range of 0.28 to 2.86 mg/L. Other studies have used influent of the same source and recorded 60.9–79.9 and 76–119 mg/L for $NH_4^+$-N and TN, respectively [22,32]. This could be explained since, as it is rural wastewater, the nitrogen contribution is higher because of the different agricultural activities in the area [33].

### 3.2. Parameters in HBCWs

Figure 2 shows the variation of the in situ parameters for the HBCWs systems. The average temperature was 15.0 °C with a minimum of 13.9 °C in winter and a maximum of 28.2 °C in summer. These data are consistent with Leiva et al. [22] who obtained similar results. They used a laboratory-scale VSSF and reported a temperature range that varied between 8.2 and 22.9 °C. There were no significant differences between the VSSF and HSSF groups ($p > 0.05$) regarding temperature. In the case of the incidence of the temperature in the biodegradation processes of organic components in anaerobic conditions, studies have concluded that there are no significant effects because bacteria are capable of performing elimination processes even at temperatures as low as 5 °C. Additionally, roots and porous media allow temperatures inside the HBCWs to be 2.3 °C above the outside temperature, thus allowing microbial activity to continue to function properly [34–36]. The pH (Figure 2a) presented values that ranged from 5.9 to 8.9, which is considered to be an optimal range for the survival of bacteria in HBCWs (4.0–9.5) [8,37]. Furthermore, higher levels of up to 10% were found in the VSSF (6.5–8.2) than in the HSSF (5.9–7.9). Likewise, the highest values of pH were registered during the autumn-winter period (up to 8.20 and 7.93 for VSSF and HSSF), and a significant difference ($p < 0.05$) was found when comparing these values to those obtained during the spring-summer period (up to 7.66 and 7.15 for VSSF and HSSF). Similar results were obtained by Marzec et al. [11]. In their study, they reported a pH range of 6.68 and 8.70 for HBCWs with a VSSF-HSSF layout that treated domestic wastewater.

Regarding conductivity, ranges of up to 1814 and 1760 µS/cm were found for VSSF 1 and VSSF 2, respectively. Conversely, HSSF 1 and 2 registered up to 1423 µS/cm and 1402 µS/cm, respectively (Figure 2b). There was a reduction of up to 34% between VSSF and HSSF, which would be explained by the absorption of micro- and macroelements and ions by plants and bacteria and their elimination through adsorption by the action of the roots of plants and sedimental suspended particles [38]. Moreover, a similar decrease was observed by Kyambadde et al. [37], who used an HBCW composed of six staggered VSSFs. In their case, a decrease of 20.72% regarding conductivity concentrations was found, going from 796.3 µS/cm to 631.5 µS/cm.

The DO concentrations in the HBCWs registered a range of 0.1–0.6 mg/L. Average values of 0.22 and 0.23 mg/L were registered for VSSF 1 and VSSF 2 (Figure 2c), respectively. No significant differences ($p > 0.05$) were found between seasons. In relation to other investigations, the concentrations in VSSF 1 and VSSF 2 were up to 89.52% lower than those reported by Sgroi et al. [39]. They recorded DO concentrations between 0.5 and 2.1 mg/L for a VSSF that treated domestic wastewater with an OLR of 40 g COD/$m^2$·d. Regarding HSSF 1 and HSSF 2 (Figure 2c), there were average concentrations of 0.4 mg/L, values up to 78.57% lower than those obtained by Caselles–Osorio et al. [40], who reported DO concentrations of 2.1 mg/L in an HSSF that treated domestic wastewater with an OLR between 4 and 5 g COD/m2·d. Higher concentrations of DO were registered during winter

with values up to 0.4 and 0.5 mg/L for VSSF 1 and VSSF 2, respectively, and 0.6 mg/L for HSSF 1 and HSSF 2. Thus, it can be determined that anaerobic conditions prevail in the HBCWs, registering concentrations < 2 mg/L of DO [41].

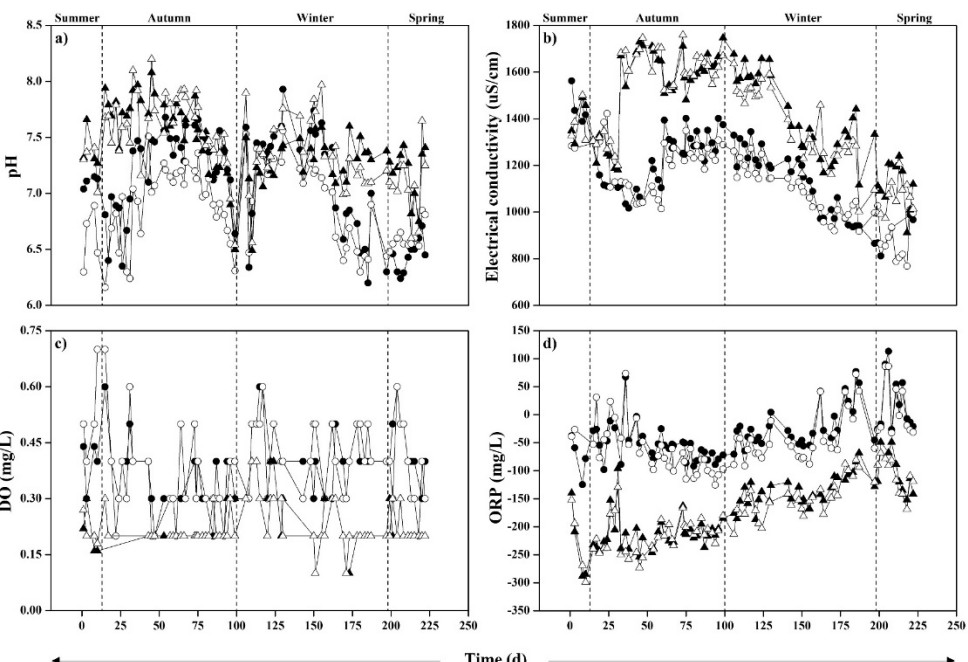

**Figure 2.** In situ parameters variation: (**a**) pH; (**b**) electrical conductivity; (**c**) dissolved oxygen; and (**d**) oxide reduction potential. Where VSSF 1 ($\triangle$); VSSF 2 ($\blacktriangle$); HSSF 1($\bigcirc$); and HSSF 2 ($\bullet$).

Regarding the measured ORP, the VSSF (Figure 2d) scored a range from 288 to −25 mV with means of −174.9 and −169.1 mV for VSSF 1 and VSSF 2, respectively. The HSSF registered values ranged from −125 to 77 mV with means of 34.7 and −35.5 mV for HSSF 1 and HSSF 2 (Figure 2d), respectively. A value higher than 100 mV is required to consider an aerobic environment, while a value lower than −100 mV indicates anaerobic conditions [42]. Consequently, the HBCWs were characterized by an anaerobic environment in both the VSSF and HSSF. The oxygen diffusion was not high enough to increase the oxygen concentration within the HBCWs and thus modify the oxidation conditions [43].

### 3.3. Concentrations and Removal Efficiencies of Organic Matter and Suspended Solids for the HBCWs System

In the VSSF, the average concentration of COD (Table 3) was kept in ranges between 70.25–92.57 mg/L with an average OLR of 23.84 and 27.81 g COD/m$^2$·d for VSSF 1 and VSSF 2, respectively. Significant differences ($p < 0.05$) were found for VSSF 1 and VSSF 2 between autumn and the rest of the seasons, causing an influent concentration of 296.94 mg/L with concentrations up to 44% higher than those in spring. In the VSSF, the average removal efficiencies for COD (Figure 3a,b) were 56.67 and 57.94% for VSSF 1 and VSSF 2, reaching higher levels of 78.41 and 82.68%, respectively, in autumn. In a recent study, Chang et al. [44] used a laboratory-scale VSSF and reached an average removal efficiency level of 59.9 and 62.8% for an influent with a COD of 288.66 mg/L.

Previous research has reported removal efficiencies that fluctuate between 41 and 92.90% [45–49]. There was only a difference of 2.21% for COD between the partially saturated VSSF system and saturated VSSF with no significant differences ($p > 0.05$). The results are consistent with the findings of Huang et al. [18]. In their study, no significant differences were found regarding the removal efficiencies of COD when using VSSF at different saturation levels (0.05, 0.30, 0.45 and 0.60 m). Therefore, it can be observed that the removal of organic matter was not affected by the alteration of the saturation level because organic matter decomposes under aerobic/anaerobic conditions as well

as by sedimentation and filtration of organic particles [50]. However, different results were reported by Bassani et al. [51]. In their study, they used an unsaturated (0.1 m) and a partially saturated VSSF (0.4 m), and significant differences ($p < 0.05$) were reported regarding the removal efficiencies of COD (87.9% and 94.3% for the unsaturated and saturated systems, respectively). These results differ from the present study since in the Bassani et al. [51] study, different HRTs (0.6–0.8 d for unsaturated VSSF and 2–2.5 d for partially saturated VSSF) were used. Therefore, a higher HRT allowed a longer period of microorganism interaction with organic matter, thus enabling better removal efficiencies of COD.

**Table 3.** Physicochemical characterization of the effluent for different CW.

| Parameter | Period | Concentration $\pm$ SD | | | |
|---|---|---|---|---|---|
| | | VSSF 1 | VSSF 2 | HSSF 1 | HSSF 2 |
| COD | Summer | 81.23 ± 25.31 | 70.25 ± 17.70 | 58.92 ± 18.23 | 57.38 ± 16.16 |
| | Autumn | 81.74 ± 18.38 | 92.57 ± 24.69 | 35.97 ± 8.84 | 41.25 ± 12.12 |
| | Winter | 84.33 ± 17.39 | 76.45 ± 8.36 | 35.25 ± 9.31 | 33.40 ± 6.11 |
| | Spring | 85.63 ± 13.78 | 84.69 ± 21.10 | 35.98 ± 3.79 | 25.07 ± 3.06 |
| TOC | Summer | 30.91 ± 4.54 | 27.29 ± 5.47 | 30.25 ± 13.44 | 13.40 ± 2.31 |
| | Autumn | 18.22 ± 4.10 | 26.96 ± 7.19 | 5.20 ± 2.01 | 5.56 ± 1.83 |
| | Winter | 18.80 ± 3.88 | 22.26 ± 2.44 | 4.57 ± 1.21 | 4.72 ± 0.86 |
| | Spring | 19.09 ± 3.07 | 24.67 ± 6.14 | 4.67 ± 0.49 | 3.54 ± 0.43 |
| TSS | Summer | 7.27 ± 2.07 | 5.41 ± 1.41 | 4.71 ± 0.42 | 3.76 ± 0.71 |
| | Autumn | 7.92 ± 2.68 | 8.33 ± 3.23 | 3.48 ± 0.53 | 3.66 ± 0.61 |
| | Winter | 5.63 ± 1.00 | 5.79 ± 1.50 | 2.93 ± 0.79 | 3.25 ± 0.94 |
| | Spring | 5.60 ± 0.10 | 6.10 ± 0.10 | 3.50 ± 0.20 | 3.27 ± 0.25 |
| VSS | Summer | 6.87 ± 1.55 | 8.19 ± 3.51 | 3.24 ± 0.85 | 3.04 ± 1.12 |
| | Autumn | 7.37 ± 2.63 | 6.03 ± 1.47 | 3.35 ± 1.91 | 2.73 ± 0.53 |
| | Winter | 4.56 ± 1.09 | 5.59 ± 1.27 | 2.38 ± 0.71 | 2.15 ± 0.77 |
| | Spring | 4.53 ± 0.06 | 4.73 ± 0.25 | 2.67 ± 0.15 | 2.73 ± 0.21 |
| | Summer | 63.50 ± 8.26 | 54.55 ± 14.36 | 13.06 ± 5.25 | 16.07 ± 7.62 |
| | Autumn | 83.25 ± 7.92 | 83.91 ± 10.18 | 18.26 ± 6.84 | 24.10 ± 10.40 |
| | Winter | 68.87 ± 15.27 | 78.89 ± 12.33 | 22.30 ± 5.32 | 34.43 ± 6.33 |
| | Spring | 33.35 ± 3.06 | 34.62 ± 7.43 | 8.86 ± 1.35 | 13.87 ± 3.09 |
| $NH_4^+$-N | Summer | 63.50 ± 8.26 | 54.55 ± 14.36 | 13.06 ± 5.25 | 16.07 ± 7.62 |
| | Autumn | 83.25 ± 7.92 | 83.91 ± 10.18 | 18.26 ± 6.84 | 24.10 ± 10.40 |
| | Winter | 68.87 ± 15.27 | 78.89 ± 12.33 | 22.30 ± 5.32 | 34.43 ± 6.33 |
| | Spring | 33.35 ± 3.06 | 34.62 ± 7.43 | 8.86 ± 1.35 | 13.87 ± 3.09 |
| $NO_2^-$-N | Summer | 0.89 ± 0.28 | 1.72 ± 0.53 | 0.06 ± 0.02 | 0.19 ± 0.09 |
| | Autumn | 0.34 ± 0.09 | 0.31 ± 0.08 | 0.21 ± 0.07 | 0.24 ± 0.06 |
| | Winter | 9.24 ± 1.97 | 4.16 ± 1.98 | 0.49 ± 0.10 | 0.32 ± 0.05 |
| | Spring | 22.79 ± 3.55 | 25.21 ± 9.38 | 1.26 ± 0.48 | 0.34 ± 0.06 |
| | Summer | 7.44 ± 2.65 | 7.76 ± 2.17 | 16.48 ± 1.25 | 8.82 ± 0.99 |
| | Autumn | 0.64 ± 0.14 | 0.69 ± 0.25 | 25.94 ± 10.11 | 16.53 ± 5.97 |
| | Winter | 2.65 ± 0.91 | 2.08 ± 0.69 | 39.37 ± 12.93 | 24.36 ± 11.15 |
| | Spring | 5.18 ± 2.75 | 5.49 ± 2.12 | 62.14 ± 1.27 | 35.63 ± 5.35 |
| TN | Summer | 82.00 ± 10.46 | 81.20 ± 17.44 | 30.40 ± 3.13 | 28.60 ± 4.34 |
| | Autumn | 96.67 ± 11.50 | 98.67 ± 11.68 | 57.00 ± 5.29 | 51.33 ± 13.32 |
| | Winter | 114.00 ± 16.37 | 113.67 ± 6.43 | 74.33 ± 7.51 | 73.33 ± 6.66 |
| | Spring | 96.00 ± 5.66 | 103.00 ± 11.31 | 70.00 ± 5.66 | 68.50 ± 9.19 |

VSSF 1: partially saturated vertical constructed wetland; VSSF 2: saturated vertical constructed wetland; HSSF 1: partially saturated horizontal constructed wetland; HSSF 2: saturated horizontal constructed wetland.

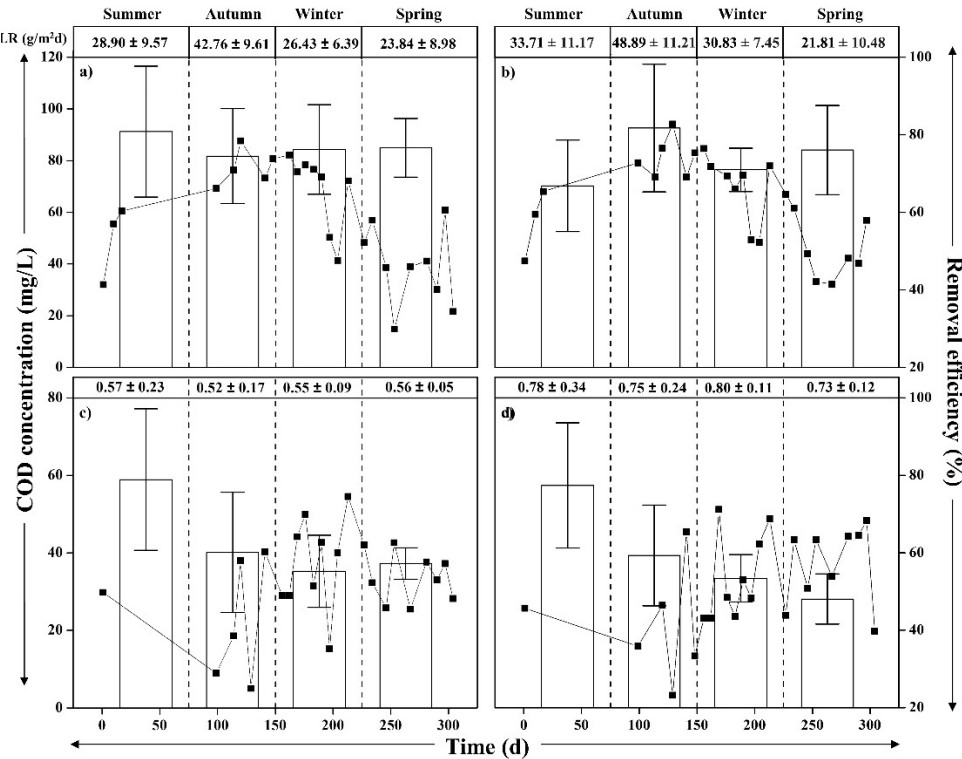

**Figure 3.** Average of effluent COD concentration (bars) and removal efficiencies (points): (**a**) VSSF 1: (**b**) VSSF 2: (**c**) HSSF 1; and (**d**) HSSF 2. OLR: organic loading rate.

HSSF registered average COD concentrations in a range of 25.07–58.92 mg/L (Table 3) with an average OLR of 0.43 and 0.59 g COD/m$^2$·d for HSSF 1 and HSSF 2, respectively. Removal efficiencies (Figure 3c,d) fluctuated between 23.2 and 74.5% with means of 49.53 and 48.56% for HSSF 1 and HSSF 2, registering removal efficiencies of up to 37% higher during winter and spring (up to 74.52 and 71.18%, respectively). These values are lower than those reported by Haddis et al. [52] who used an HSSF that treated municipal wastewater and recorded an average removal efficiency of 65%. Other studies have reported organic matter removal efficiencies in HSSF that range between 33.60 and 90.22% [53,54]. A difference of 1.90% for COD was observed between the partially saturated HSSF and the saturated HSSF; therefore, no significant differences ($p > 0.05$) were found. These results differ from those reported by García et al. [15]. In their study, significant differences ($p < 0.05$) were found for COD when using a saturated HSSF (0.50 m) and a partially saturated HSSF (0.27 m). Furthermore, in their study, the partially saturated system registered removal efficiencies of up to 83% versus removal efficiencies of 65% for the saturated system. Similarly, Al-Ajalin et al. [30] reported significant differences ($p < 0.05$) using an HSSF with two saturation levels (0.35 and 0.45 m) with higher removal efficiencies of COD in the system with a lower saturation level (up to 96.94%). On the other hand, Benvenuti et al. [55] indicated that a DO concentration higher than 1.5 mg/L is needed to allow aerobic microbial metabolism. Therefore, it can be expected that in a VSSF and HSSF that included DO < 1.5 mg/L concentrations and a reductive environment (between −288 and 77 mV), both in the partially saturated and saturated systems, organic matter was removed mainly by anaerobic processes [56], thus explaining that there were no significant differences between the saturation levels.

The removal efficiencies of COD in the HBCWs were 79.01 for HB 1 and 79.79% for HB 2. There were no significant differences ($p > 0.05$) between HB 1 and HB 2 regarding removal efficiencies. In a recent study by Kraiem et al. [57], removal efficiencies of COD of 86% were reported when using an HBCW with a VSSF-HSSF layout that treated rural wastewater. In both VSSF and HSSF, higher removal efficiencies of COD were achieved during cold seasons (autumn-winter), thus concluding that temperature did not affect the removal processes of organic matter. Likewise, Rozema et al. [58] evaluated the effect of

temperature in a VSSF, concluding that cold weather did not affect the VSSF treatment efficiencies; furthermore, efficiencies reached up to 99% for COD.

These were removed mainly by physical mechanisms such as sedimentation and filtration in relation to the TSS. Filtration occurred in the roots and stems of macrophyte plants and in the support medium [29]. Additionally, VSSF 1 and VSSF 2 presented removal efficiencies of 83.55 and 82.24%, respectively, and no significant differences were found ($p > 0.05$) between seasons and amid the partially saturated VSSF and the saturated VSSF. The values reported are higher than those reported by Abdelhakeem et al. [59]. They reported an average removal efficiency of 75% for a laboratory-scale VSSF that used municipal wastewater. HSSF 1 and HSSF 2 recorded removal efficiencies of 57.24 and 50.60%, respectively. Jácome et al. [60] registered removal efficiencies of 77% in an HSSF that treated municipal wastewater with concentrations of TSS in the influent of 41–48 mg/L (up to 22% lower than the current study). There were no significant differences ($p > 0.05$) between seasons, but there was a significant difference ($p < 0.05$) of 11.60% between the partially saturated HSSF and saturated HSSF systems, mainly because of the increase in the saturation level, which in turn generated a free flow in the upper part of the support medium. As a consequence, the removal capacity of plants decreased when removing TSS through the roots [61]. Additionally, HB 1 and HB 2 reached average removal efficiencies of 89.81 and 90.53%, respectively. Moreover, there were no significant differences ($p > 0.05$) between HB 1 and HB 2 regarding the removal efficiencies of TSS. The values reported are consistent with previous research that registered average removal efficiencies of 94–96% [11,12].

### 3.4. Concentrations and Removal Efficiencies of Nitrogen for the HBCWs System

VSSF 1 (Figure 4a) registered lower concentrations of $NH_4^+$-N during spring, at 29.97 mg/L. On the other hand, VSSF 2 (Figure 4b) $NH_4^+$-N concentrations fluctuated between 28.53 and 96.18 mg/L with lower average concentrations (Table 3) during spring (34.62 mg/L). Additionally, in HSSF 1 (Figure 5c), the minimum concentration of $NH_4^+$-N was recorded during summer (6.92 mg/L). In the case of HSSF 2 (Figure 4d), the $NH_4^+$-N remained between 7.61 and 42.67 mg/L, recording the lowest concentration during spring (13.87 mg/L).

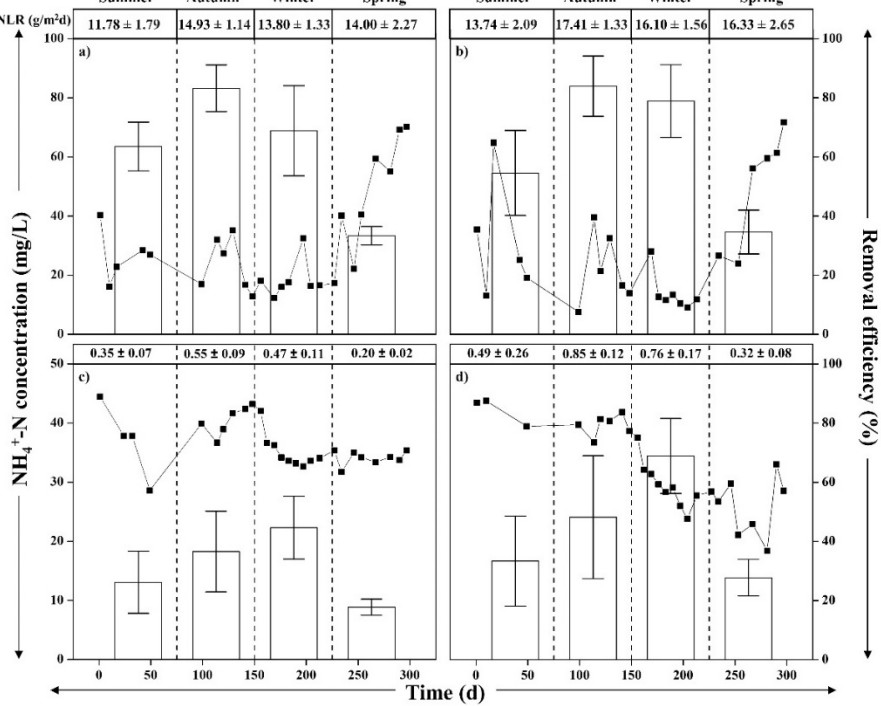

**Figure 4.** Effluent average $NH_4^+$-N concentration (bars) and removal efficiencies (points): (**a**) VSSF 1; (**b**) VSSF 2; (**c**) HSSF 1; and (**d**) HSSF 2. NLR: nitrogen loading rate.

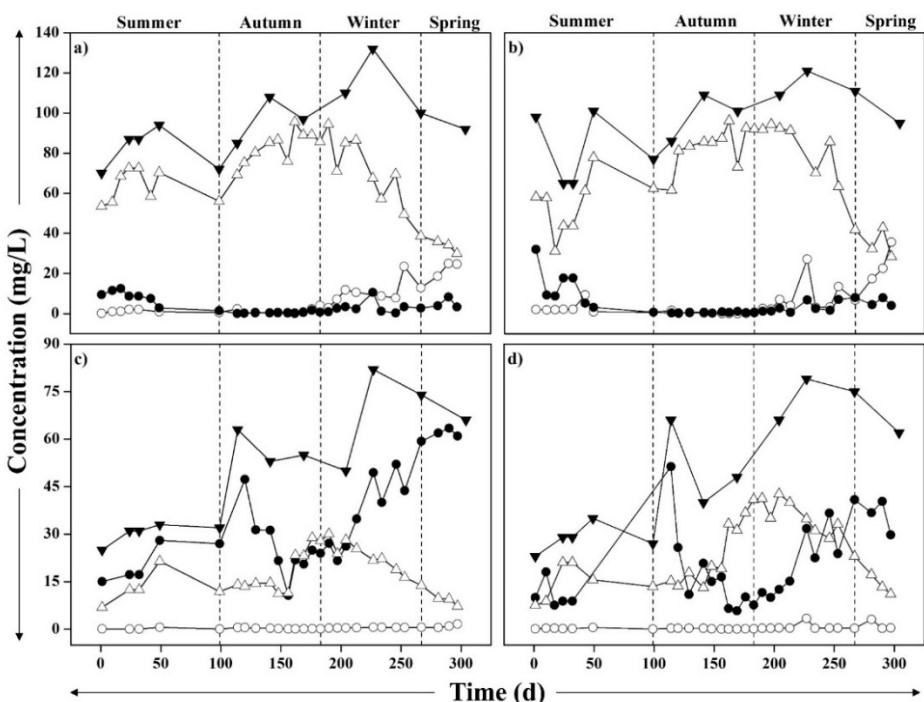

**Figure 5.** Concentrations of different forms of nitrogen (mg/L) $NO_2^--N$ ($\bigcirc$), $NO_3^--N$ ($\bullet$), $NH_4^+-N$ ($\triangle$) and TN ($\blacktriangledown$): (**a**) VSSF 1; (**b**) VSSF 2; (**c**) HSSF 1; and (**d**) HSSF 2.

For the removal efficiencies of $NH_4^+-N$, VSSF 1 (Figure 4a) registered a range of 12.28–70.22% with an average of 34.47%. For VSSF 2, a range of 7.52–71.66% (Figure 4b) was recorded with higher efficiencies during spring (up to 71.66%) and an average of 33.40%. Furthermore, significant differences ($p < 0.05$) were found between VSSF 1 and VSSF in spring and the rest of the seasons. In a similar study, Abdelhakeem et al. [59] reported removal efficiencies of $NH_4^+-N$ of 36% for a laboratory-scale VSSF that registered an effluent with $NH_4^+-N$ concentrations of 21 mg/L (37.5% lower than the average concentrations of this study). Moreover, several studies have reported removal efficiencies of $NH_4^+-N$ that have reached values up to 88.0–97.8% [62–64]. Additionally, a difference of only 3.01% for $NH_4^+-N$ was found between the partially saturated VSSF and the saturated VSSF; thus, no significant differences ($p > 0.05$) were reported, and nitrification was not affected when modifying the saturation level. In contrast, Huang et al. [18] reported significant differences ($p < 0.05$) concerning the removal efficiencies of $NH_4^+-N$ when using different saturation levels (0.05, 0.30, 0.45 and 0.60 m). Furthermore, when changing the saturation level from 0.45 to 0.60 m, a decrease of 12.71% in removal efficiencies was reported. This can be explained by the difficulty of diffusing oxygen at 0.45 m of saturation.

HSSF registered average removal efficiencies regarding $NH_4^+-N$ of 73.95–65.93% for HSSF 1 and HSSF 2. For HSSF 1 (Figure 4c), removal efficiencies of $NH_4^+-N$ higher than 60% were recorded, reaching 88.98% in summer. Similarly, HSSF 2 (Figure 4d) registered removal efficiencies in the range of 36.82–87.49%, reaching 87.49% in summer. Furthermore, significant differences ($p < 0.05$) were found for HSSF 1 and HSSF 2 during the autumn-winter period. The values reported are higher than those obtained by Zurita et al. [31]. In their study, HSSF was used, reaching $NH_4^+-N$ removal efficiencies of 45.8–48.6%. Additionally, for the partially saturated and saturated HSSFs, although there was a difference of 10.85% in $NH_4^+-N$, there were no significant differences ($p < 0.05$). Sanchez-Ramos et al. [17] obtained similar results when using HSSF at different saturation levels (partially saturated (0.27 m) and saturated (0.50 m)). They reported significant differences ($p < 0.05$) in $NH_4^+-N$ since the partially saturated system registered $NH_4^+-N$ removal efficiencies up to 56% higher than those of the saturated system. Moreover, a higher level of saturation generated changes at the redox state level, creating more reducing conditions and changing

oxygen availability as a result of a shorter diffusion of oxygen from the surface and the air-water interface, which are fundamental in the transformation processes of nitrogen components [65].

DO concentrations (<2 mg/L), particularly in the VSSF, would be caused by the oxygen demands for the organic matter oxidation of (min 1.5 mg/L DO), generating a competition of the DO, thus decreasing the ammonium-oxidizing bacteria's performance and, as a consequence, nitrification [66,67]. According to Liu et al. [67], DO concentrations higher than 1.5 mg/L are needed for nitrification to occur. The previously mentioned factors would help explain the VSSF's lower performance in removing $NH_4^+$-N (34.47% for VSSF vs. 73.95% for HSSF), owing to lower concentrations of available DO (0.2 mg/L for VSSF vs. 0.4 mg/L for HSSF). In addition, different studies have suggested that temperature affects the nitrification process with the optimal temperature ranging from 16 to 32 °C [54,68,69]. With regard to VSSF and HSSF, the removal efficiencies of $NH_4^+$-N were higher, reaching 70.22 and 37.00% in the warmer seasons (spring-summer), respectively. In their study, Hua et al. [68] reported a decrease in removal efficiencies of $NH_4^+$-N of up to 65% to 20% from summer to winter, caused by the reduction of activity and proliferation of nitrification microorganisms amid a decrease in the temperature [70].

The $NO_2^-$-N registered values ranged from 0.26–23.53 mg/L for VSSF 1 (Figure 5a) and 0.23–35.63 mg/L for VSSF 2 (Figure 5b). In the case of HSSF 1 (Figure 5c), $NO_2^-$-N concentrations lower than 2 mg/L were recorded throughout the monitoring period with values ranging between 0.035 and1.61 mg/L. On the other hand, HSSF 2 (Figure 5d) had an average value for $NO_2^-$-N of 0.019 mg/L with a range of 0.01–3.35 mg/L. The $NO_3^-$-N in VSSF 1 (Figure 5a) remained at values lower than 13 mg/L, in a range of 0.07–12.56 mg/L. In the case of VSSF 2 (Figure 5b), in general, $NO_3^-$-N concentrations remained lower than 5 mg/L, except in summer, where concentrations up to 32 mg/L were recorded. The HSSF 1 (Figure 5c) registered concentrations of $NO_3^-$-N ranged between 10.65 and 63.48 mg/L. On the other hand, HSSF 2 (Figure 5d) recorded $NO_3^-$-N values that reached 51.32 mg/L in autumn. Furthermore, in the VSSF, there were lower concentrations of $NO_3^-$-N than in the HSSF (up to 97% in autumn), which could have been removed through denitrification under anaerobic conditions [66]. Alternatively, the decrease in $NO_3^-$-N could also be explained by simultaneous nitrification/denitrification (SND) processes [71]. This happens as the biofilm is formed in the support medium; thus, an anaerobic microenvironment is created inside, and an aerobic microenvironment is created outside, allowing the coexistence of facultative aerobic and anaerobic microorganisms [72,73].

Conversely, the highest concentrations of $NO_3^-$-N in HSSF (up to 63.48 mg/L) could be caused by the suppression of denitrification, owing to the lack of an organic carbon source, since it significantly affects the efficiency of denitrification, causing an accumulation of $NO_3^-$-N in the medium [8,74]. Ding et al. [74] used different C/N relationships (0, 2, 4, 6 and 9) and established that the optimal removal appeared when increasing the C/N relationship to 9, where there were concentrations of $NO_3^-$-N in the effluent of only 0.03 mg/L for an influent of 19.98 mg/L.

In this study, the lowest concentrations of C/N were found in the HSSF. Furthermore, HSSF 1 registered a C/N relation of 0.8 during the monitoring period, and HSSF 2 registered C/N relations of 1.0, 0.8, 0.7 and 0.7 during summer, autumn, winter, and spring, respectively, thus explaining the accumulation of $NO_3^-$-N in these systems.

In VSSF 1 (Figure 5a), the TN remained in the range of 70–132 mg/L, and in winter, lower concentrations (70 mg/L) were registered. On the other hand, in VSSF 2 (Figure 5b), TN concentrations fluctuated between 65.00–121.00 mg/L, registering a similar concentration in summer (65.00 mg/L). Additionally, in HSSF 1 (Figure 5c) and HSSF 2 (Figure 5d), the lowest concentrations of TN were recorded in summer, at 33 and 23 mg/L, respectively. Regarding removal efficiencies, VSSF recorded average efficiencies of 22.64 and 18.78% for VSSF 1 and VSSF 2, respectively. These values were up to 62.44% lower than those reported by Zurita et al. [31], who used a VSSF that treated domestic wastewater and reported removal efficiencies of 50% for an influent with concentrations of TN of 28.7 mg/L (up to

77% lower than the concentrations of TN in this study). Additionally, significant differences ($p < 0.05$) were found between the partially saturated VSSF and the saturated VSS with a difference of 17.05% for TN. This could be caused by changes in the saturation levels creating changes at the DO level (0.62 to 2.2 mg/L) and ORP ($-5.39$ to 90.93 mV), causing the different microbial communities that act simultaneously in the medium to be affected, thus altering nitrogen removal processes such as nitrification and denitrification [75–77]. In the case of the HSSF, average removal efficiencies of TN of 37.03 and 39.40% were recorded for HSSF 1 and HSSF 2, respectively. In addition, no significant differences ($p > 0.05$) were found between the systems. In a recent study, Jácome et al. [60] used HSSF and established a TN removal efficiency of 37%. On the other hand, Zhu et al. [78] reported removal efficiencies of TN that reached 89.9%; however, this efficiency was due to the TN concentration in the influent of 40 mg/L, which was up to 68% lower than the average concentration of TN in this study (126 mg/L).

In HBCWs, the removal efficiencies of $NH_4^+$-N reached 83.50 for HB 1 and 77.28% for HB 2. Saeed et al. [14], who used partially saturated HBCWs, obtained similar results. They established removal efficiencies of $NH_4^+$-N of 81.1%. On the other hand, these values are lower than those reported by Ávila et al. [79], who registered values of up to 94%. Regarding TN, average removal efficiencies of 55.12 for HB 1 and 57.40% for HB 2 were reported. Moreover, Nguyen et al. [13] reported lower values than those previously mentioned, where removal efficiencies of 79% were reported using municipal wastewater for HBCWs with a layout of VSSF-HSSF.

### 3.5. In Situ Spatial Variations and Physical-Chemical Parameters

Table 4 shows the in situ spatial variations and physicochemical parameters. In the case of the COD, it can be observed that during 2019 (data with influent concentrations), the greatest decrease in concentrations occurred in the upper zone with reductions of 58.64 and 53.54 mg/L for VSSF 1 and VSSF 2, namely, removal efficiencies of 34 and 31%, respectively. As the zone being measured decreased, removal efficiencies declined since only 16% was recorded in the lower zone. In a study conducted by Yi et al. [80], it was established that COD concentrations were mainly removed at the entrance with a removal efficiency of 64.23% in the upper zone. Previous research has reported that the anaerobic routes of organic matter removal are slower than aerobic routes [19], which would explain the decrease in the removal efficiency in the lower zones when dealing with systems with a low HRT.

**Table 4.** Spatial variations of in situ and physicochemical parameters.

| Year | CW | Location | Concentration (mg/L) | | | | | | |
|---|---|---|---|---|---|---|---|---|---|
| | | | COD (mg/L) | $NH_4^+$-N (mg/L) | Temperature (°C) | pH | ORP (mV) | DO (mg/L) | EC (µS/cm) |
| 2018 | VSSF1 | Upper | - | - | 10.9 ± 1.3 | 7.2 ± 0.2 | −166.9 ± 44.6 | 0.5 ± 0.2 | 1247.3 ± 393.1 |
| | | Middle | - | - | 10.7 ± 1.4 | 7.1 ± 0.1 | −194.1 ± 47.5 | 0.3 ± 0.1 | 1232.2 ± 360.5 |
| | | Bottom | - | - | 10.6 ± 1.2 | 7.1 ± 0.1 | −162.4 ± 127.9 | 0.3 ± 0.0 | 1034.0 ± 79.2 |
| | VSSF2 | Upper | - | - | 10.9 ± 0.5 | 7.4 ± 0.2 | −162.9 ± 99.3 | 0.4 ± 0.2 | 1170.3 ± 167.2 |
| | | Middle | - | - | 10.5 ± 0.6 | 7.2 ± 0.3 | −164.3 ± 127.1 | 0.3 ± 0.1 | 1139.3 ± 111.1 |
| | | Bottom | - | - | 10.0 ± 0.5 | 7.2 ± 0.2 | −193.2 ± 83.2 | 0.2 ± 0.1 | 1031.3 ± 122.7 |
| 2019 | VSSF1 | Upper | 111.9 ± 14.5 | 65.8 ± 12.6 | 11.1 ± 1.1 | 7.4 ± 0.2 | −112.4 ± 24.8 | 0.3 ± 0.1 | 1339.0 ± 95.3 |
| | | Middle | 104.7 ± 19.4 | 68.9 ± 13.5 | 11.0 ± 1.2 | 7.4 ± 0.2 | −135.5 ± 36.6 | 0.3 ± 0.1 | 1341.0 ± 98.7 |
| | | Bottom | 87.6 ± 18.9 | 58.8 ± 13.2 | 10.9 ± 1.2 | 7.3 ± 0.1 | −141.2 ± 33.3 | 0.3 ± 0.1 | 1332.0 ± 111.2 |
| | VSSF2 | Upper | 117.0 ± 17.0 | 73.7 ± 7.1 | 11.2 ± 1.1 | 7.4 ± 0.2 | −133.6 ± 14.9 | 0.3 ± 0.1 | 1361.7 ± 65.6 |
| | | Middle | 92.2 ± 20.2 | 62.9 ± 17.7 | 11.1 ± 1.2 | 7.4 ± 0.2 | −154.7 ± 11.9 | 0.3 ± 0.1 | 1354.0 ± 54.6 |
| | | Bottom | 77.2 ± 20.2 | 55.9 ± 15.3 | 10.9 ± 1.2 | 7.1 ± 0.0 | −153.9 ± 11.9 | 0.3 ± 0.1 | 1182.3 ± 38.8 |

Similarly, for $NH_4^+$-N, the highest concentration decline was recorded in the upper zone with 31.43 mg/L for VSSF 1 and 23.53 mg/L for VSSF 2, namely, removal efficiencies of 32% and 24%, respectively. Sanchez-Ramos et al. [17] found that aerobic bacteria are

distributed in the first centimeters of the wetland, thus taking advantage of the atmospheric diffusion of oxygen. Therefore, it can be expected in his study that nitrification bacteria were developed in the first centimeters of the wetland, thus explaining the removal rates for $NH_4^+$-N despite the ORP and DO values obtained, which hinder the nitrification process.

If we consider the in situ parameters, the pH presented a decrease of 0.1 pH units when the measurement zone descended vertically with a range between 7.1–7.4 for VSSF 1 and VSSF 2, and no significant differences ($p > 0.05$) were found. Thus, the pH was not vertically affected. Regarding DO, during 2018, there were higher concentrations in the upper zone with 0.5 mg/L for VSSF 1 and 0.4 mg/L for VSSF 2. Moreover, as the measured zone descended, the DO concentrations decreased, reaching 0.3 mg/L for VSSF 1 and 0.2 mg/L for VSSF 2. However, during 2019, the DO concentration remained consistent as the vertical measurement decreased. This vertical decline could be caused by the removal process of organic matter and nitrogen transformation, which in turn would cause a rapid consumption of DO through aerobic respiration and chemical oxidation [42]. In the case of the ORP, a range of values registered from −112.4 mV in the upper zone to −162.4 mV in the lower zone of VSSF 1. On the other hand, VSSF 2 registered a range from −133.6 mV in the upper zone to −193.2 mV in the lower zone. This decline could be caused by the oxygen diffusion mainly in the upper zone of the gravel beds, decreasing as the depth increases, thus creating more reducing conditions [15].

### 4. Conclusions

In this study, the medium saturation level effect in the VSSF and HSSF does not show significant difference when the COD removal efficiencies are considered. However, when the saturation level of the medium increased, the TSS removal efficiency decreased in 11% in the HSSF. Moreover, this also meant a 17% decrease in the average removal efficiencies of TN in VSSF and a 10% decrease for $NH_4^+$-N at HSSF. Due to this, we conclude that when designing HBCWs, it is essential to consider the medium saturation level because it affects the transformation and/or removal of wastewater's components to be treated. Even more so, the increase of saturation level in HBCWs reduces the transformation of the $NH_4^+$-N. This research shows the possibility of optimizing the transformation of nitrogen with partially saturated hybrids constructed wetlands.

As future prospects for this work, it is important to enhance a mass balance of the different forms of nitrogen ($NH_4^+$-N, $NO_3^-$-N $NO_2^-$-N), considering the microbiological transformations and plant assimilation. In this same line, in future research it would be important to evaluate the gas production and emissions such as $CH_4$, $CO_2$ or $N_2O$ due to the use of partially saturated wetlands and also optimize the system to control said emissions.

**Author Contributions:** Conceptualization, D.L. and G.V.; methodology, D.L. and J.C.; software, G.G.; validation, G.V., D.L. and J.C.; formal analysis, J.C. and D.L.; investigation, J.C. and D.L.; resources, G.V.; data curation, J.C. and G.G.; writing—original draft preparation, J.C. and G.G.; writing—review and editing, G.V., D.L. and G.G.; visualization, G.G.; supervision, G.V.; project administration, G.V. and D.L.; funding acquisition, G.V. All authors have read and agreed to the published version of the manuscript.

**Funding:** This research was funded by ANID/FONDAP/15130015 and Grant No. 3170295 from ANID-FONDECYT (Chile).

**Conflicts of Interest:** The authors declare no conflict of interest.

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
