# Peer review of "Seasonal Enhancement of Nitrogen Removal on Domestic Wastewater Treatment Performance by Partially Saturated and Saturated Hybrid Constructed Wetland"

_water, doi:10.3390/w14071089_

Round 1
Reviewer 1 Report
In this work, partially saturated and saturated hybrid constructed wetlands were utilized in enhancing nitrogen removal from real wastewater influent. The manuscript is well organized and results interpretation is often reasonable. However, at some points, the submitted manuscript leaves something to be desired. Hence, the manuscript could be recommended for publication after considering the following comments:
Comment #1: In abstract, more qualitative findings should be provided.
Comment #2: Introduction should be broadened, considering highlighting the knowledge gaps in the literature and the novelty aspects of this study.
Comment #3: The main aim and objectives of the study should be clearly stated at the end of introduction section.
Comment #4: Why did the authors consider HBs as the acronym for hybrid constructed wetlands, instead of HBCWs for instance?
Comment #5: The manuscript still needs some English proofreading work, in terms of choosing the appropriate expressions.
Author Response
Dear Reviewer #1,
The answers are in the attached file.
Best regards,
Gladys

Reviewer 2 Report
The paper by Contreras and co-authors deals with an important topic, i.e. the use of constructed wetlands for nitrogen removal in domestic wastewaters, however I cannot express positively on it. My main concerns are:
- as it stands, the paper reads like a regional technical report. I think there need to be a partial rewrite of the introduction, discussion, and conclusions to convert the manuscript into a more interesting scientific paper for an international audience. Studies dealing with the performance of CW treating wastewaters are quite common in the literature. The authors should stress more the novelty of their work.
- A deeper and speculative interpretation of the presented data is needed, in a way to better evaluate both the scientific level of the presented outcomes and also their spendibility for practical application in the construction and management of CW.
Specific comments:
- In the Abstract, the authors did not even mention where the study was conducted (i.e. study area).
- line 28: Define VSSF and HSSF the first time you mention them in the text.
- line 31: I think the term "elimination" is not completely appropriate. Nitrification supports the oxidation of ammonia to nitrate but not the complete removal (=elimination) of nitrogen from the system.
- line 37: saturation of what? Oxygen saturation? Not clear, please reformulate. In general, the Introduction needs to be more focused on how saturation may affect the N cycling processes and thus the treatment performance of CW.
- lines 64-65: Sentence not clear, please reformulate.
- line 75: Some information about macrophytes in VSSF and HSSF are needed. For example, density or biomass per m2 are similar in VSSF and HSSF?
- line 76: What is the unit of measure here? Maybe L/day instead of L?
- line 107: The authors need to describe the procedure used to estimate the pollutant removal efficiencies.
- line 156: These data reflect the climatic region where the wetlands is located.
- line 168: Why was conductivity significant lower in spring-summer period?
- Conclusion section: there is only a repetition of the main results, consider shorten this paragraph. Concluding remarks regarding the specific goal and hypothesis of the study are needed to provide the reader with a take home message and a final summary of ideas, management implications and future studies.
Author Response
Dear Reviewer #2
The answers are in the attached file.
With my best regards,
Gladys

Author Response
Dear Reviewer #3,
The answers are in the attached file.
Best regards,
Gladys
